# Effect of Textile Wastewater Secondary Effluent on UF Membrane Characteristics

**DOI:** 10.3390/polym14102035

**Published:** 2022-05-16

**Authors:** Iva Ćurić, Davor Dolar, Josip Horvat, Katia Grgić

**Affiliations:** 1Department of Physical Chemistry, Faculty of Chemical Engineering and Technology, University of Zagreb, HR-10000 Zagreb, Croatia; icuric@fkit.hr (I.Ć.); jhorvat@fkit.hr (J.H.); 2Department of Textile Chemistry and Ecology, Faculty of Textile Technology, University of Zagreb, HR-10000 Zagreb, Croatia; katia.grgic@ttf.unizg.hr

**Keywords:** ultrafiltration, membrane fouling, textile wastewater, membrane characteristics

## Abstract

The influence of textile wastewater (TWW) secondary effluent on ultrafiltration (UF) membrane characteristics was investigated. TWW treated with a membrane bioreactor was subjected to four commercial UF membranes (2, 3, 5, and 10 kDa). Both the pristine membranes and the membranes after TWW secondary effluent treatment were characterized. Surface roughness, microscopic analysis of the membrane surface and cross-section, zeta potential, contact angle, membrane composition, and membrane flux were compared. After treatment of secondary effluent, the zeta potential decreased for 5 and 10 kDa membranes, while the contact angle and surface roughness increased for all investigated membranes. In addition, a fouling layer formed on all membranes, and new interactions with pollutants and membranes were confirmed. Membranes with larger pores (5 and 10 kDa) showed a greater decrease in permeate flux during treatment. Detailed analysis showed variations in membrane characteristics after TWW secondary effluent treatment, indicating the stability of the membranes used.

## 1. Introduction

Membrane processes have been widely used in the removal of pollutants from wastewater and the reuse of water in industry [1]. Using the Fuzzy-Delphi approach, these processes are recognized as Best Available Techniques (BAT) according to high technical, environmental, economic, and social criteria [2]. In recent years, for example, pressure membrane driving processes have been widely used in the textile industry for the treatment of their heavily polluted wastewater generated during wet processing [3,4,5]. Microfiltration (MF) and ultrafiltration (UF) membranes are usually used as pretreatment, and UF can be used as treatment for the reuse of wastewater in minor processes (washing, etc.) with moderate dye rejection [6]. These processes provide good removal of colloidal dyes and suspended solids (SS) from wastewater, which can cause irreversible fouling of membranes, especially in nanofiltration (NF) and reverse osmosis (RO) [7]. Due to the characteristics of the wastewater itself, the treatment of textile wastewater (TWW) requires the use of membranes that have high stability. Typical characteristics of TWW are high color, conductivity, SS, chemical oxygen demand (COD), biochemical oxygen demand (BOD), etc.

In order to determine the sufficient stability of membranes, it is necessary to determine the characteristics of membranes. The most important characteristics of membranes are their chemical composition, hydrophilicity/hydrophobicity, charge, and morphology (surface roughness, pore size, etc.) [8]. Contact angle analysis, Atomic Force Microscopy (AFM), Scanning Electron Microscopy (SEM), zeta potential, and Fourier Transform Infrared Spectroscopy (FTIR) are the main techniques used to characterize membranes.

In their study, Zhank et al. [9] showed changes in zeta potential for membrane with a pore size of 0.2 µm during the treatment of pickling wastewater from the steel industry. The zeta potential of the fouled membranes was lower compared to that of new membrane from pH 3.48 to 7.69 and higher. In addition, they showed that the contact angle, or hydrophilicity, decreased after treatment, which reduced the membrane’s antifouling ability. In their study, Thuvander et al. [10] reported decomposition on the surface of the UF membrane visible on SEM after treatment of process water from a pulp mill. Shon et al. [11] investigated contact angles after UF treatment of biologically treated sewage effluent. They showed that the contact angle of UF membranes increased after treatment, indicating higher hydrophobicity. Moreover, FTIR analysis carried out in the same study showed new peaks in the FTIR spectrum, confirming a change in the surface composition of the UF membrane.

Fouling is an inherent phenomenon and a major problem in membrane processes, affecting membrane performance and increasing treatment operating costs. In seeking to extend membrane life the characterization of membranes is critical to studying the life cycle of a membrane, as it allows fouling to be predicted and reduced. Previous studies on TWW treatment and fouling monitoring have focused on flux changes, as this is one of the simplest fouling indicators and is easy to measure [2,12,13]. To date, no study has addressed the detailed characterization of UF membranes. Therefore, the goal of this study is the detailed characterization of UF membranes after treatment of real TWW secondary effluent. The pristine membranes and the membranes after TWW treatment were subjected to compositional, morphological, structural, and performance characterization, i.e., the UF membranes (2, 3, 5, and 10 kDa) were analyzed by AFM, SEM, zeta potential, contact angle, FTIR, and membrane flux. The presented study can provide crucial insight into membrane changes during wastewater treatment, and eventually help to reduce the costs of TWW reuse with membrane processes.

## 2. Materials and Methods

### 2.1. Membranes

Four UF membranes were used in this work. The main characteristics provided by the manufacturers are listed in Table 1. The pristine membranes were characterized after washing the conserving agent with 3 L of demineralized water and stabilizing the flux at the working pressures provided in Table 1. The same membranes were used for TWW treatment. All membranes were dried in an oven at 35 °C for up to 15 h to remove 100% of the water residues in the pores and on the surface, as required for further characterization.

### 2.2. Textile Wastewater Treatment

The TWW used in this study was pretreated with an MBR and separately subjected to four UF processes. The MBR treatment was carried out for 4 weeks, and the physicochemical parameters of the MBR permeate were determined daily: pH, conductivity (*κ*), turbidity, SS, COD, total organic carbon (TOC), total nitrogen (TN_b_), color expressed by spectral absorption coefficient (SAC) at wavelengths of 436, 525, and 620 nm, and the concentrations of Cl^−^, SO_4_^2−^, Na^+^, K^+^, Mg^2+^, and Ca^2+^ ions. The methods and apparatus were as described in detail in the studies by Ćurić et al. [14,15]. The MBR permeate was subjected to each individual UF membrane.

The UF laboratory setup was performed in batch circulation mode, described in detail in the study by Racar et al. [16], with a membrane area of 0.00138 m^2^ and crossflow velocity of 0.75 m s^−1^. The final step was washing the membrane with demineralized water for 30 min and drying, as described in Section 2.1 of the above study. The flux was monitored with a technical balance (KERN 440–35A, Balingen, Germany), while the computer recorded the mass every 10 s.

### 2.3. Membrane Characterization

The contact angle was determined by the sessile drop method using a DataPhysics OCA 20 goniometer (Filderstadt, Germany). A drop of MilliQ water was formed manually using a micrometer doser, and the drop image was recorded using the camera built into the instrument. The volume of the drop was 2 μL at room temperature (23.5 ± 1.0 °C). To minimize experimental error, contact angles were measured at four random locations for each sample and the average value was reported (*N* = 4).

A Bruker Vertex 70 (Ettlingen, Germany) FTIR analyzer with a Platinum ATR single reflection diamond (*n* = 2.4) crystal-based module in the mid IR range (400–4500 cm^−1^) was used to characterize the surface of the membranes. The membrane spectra were recorded with a resolution of 4 cm^−1^ and 32 scans.

The surface and cross section of the membranes were visually analyzed using SEM (Tescan Vega III Easyprobe, Kohoutovice, Czech Republic) at 10 kV and a magnification of 2500×. The samples had been previously coated with gold and palladium.

The zeta potential of the membranes was determined using the SurPASS Electrokinetic Analyzer-type A device from Anton Paar (Graz, Austria). Using the Adjustable Gap Cell, the 20 mm × 10 mm sample was fixed to the sample holder with double-sided adhesive tape. The pH dependence of the zeta potential was determined in the range of pH 2.5–9.

Atomic force microscopy characterization was performed using CoreAFM (Nanosurf, Liestal, Switzerland) under ambient conditions. Non-contact (tapping) mode was used for image acquisition. The Tap300Al-G probe with a nominal resonant frequency of 300 kHz and a tip radius of less than 10 nm proved to be the best solution for this characterization. The scan parameters were initialized with a nominal value of 35 or 55 nN contact force and an acquisition time of 0.78 s on an area of 10 × 10 µm. The images were processed using the Gwyddion program. The AFM images were used to calculate the surface roughness parameters of the membranes, expressed by the mean roughness (*S*_a_) and the root mean square of the Z data (*S*_q_).

## 3. Results and Discussion

### 3.1. TWW Secondary Effluent Characteristics

Biological treatment of wastewater, including TWW, produces effluents that contain effluent organic matter (EfOM), which is of great concern to human health and the environment. In addition, EfOM consists of natural organic matter (NOM) and soluble microbial products (SMPs). EfOM is mainly present as a soluble fraction and is therefore difficult to remove. In the case of TWW, this is related to dye residues with additives (chromophores and complex polyaromatic structures), which are difficult to biodegrade [17]. In the case of NOM, it can be related to the use of drinking water in the production process in the factory where TWW was sampled, while SMPs result entirely from biologically treated wastewater. NOM is problematic for membrane processes because it is the main source of organic fouling on membranes, i.e., it can physically interact with and chemically degrade the membrane [18]. This study was conducted with TWW from the textile industry, in which the most common additional pollutants are dyes, detergents, and salts (containing Cl^−^, SO_4_^2−^, Ca^2+^, K^+^, Mg^2+^, Na^+^, and caustic soda) [18]. The concentrations of these pollutants are shown in Table 2. It can be seen that the concentrations of most of these pollutants are relatively high, i.e., up to 1228.6 mg L^−1^ for sulphate.

### 3.2. Membrane Surface Charge

Membrane charge can provide information about separation when the MWCO of the membranes used is much higher than the components found in the TWW. An example of this is the humic and fulvic acids present in drinking water, as they are part of the EfOM and have a molecular weight between 10^3^ and 10^6^ Da [11], which corresponds to the MWCO of the membranes used. The zeta potential can be used to predict whether there are any electrostatic interactions between charged solutes and the charged membrane surface [19].

Figure 1 shows the zeta potential of all pristine membranes and of the membranes after treatment of TWW secondary effluent. From Figure 1a,b, it can be seen that the zeta potential of the 2 and 3 kDa membranes, both PA, have similar curves. It is interesting to note that neither membrane has an isoelectric point, i.e., the zeta potential is negative over the entire range of measurements, with the lowest zeta potential at about −50 mV at a maximum pH of 9.3. The PES/PSf (5 and 10 kDa) pristine membranes showed a different zeta potential curve compared to the PA membranes. The first difference was the presence of an isoelectric point at pH 3.0, and the second was the higher (more positive) zeta potential of the PES/PSf membrane compared to the PA membranes. For 5 and 10 kDa membranes, the lowest zeta potential was −30 mV (pH = 9.5) and −12 mV (pH = 9.0), respectively. It can thus be concluded that the polymer material affects the zeta potential of the membranes. Similar values for PES membranes have been confirmed in the study of Manawi et al. [20], in which the authors indicated that the increase in the surface charge of the membrane, together with increased pH, is related to the adsorption of chloride ions from the background electrolyte solution (KCl) on the membrane surface. Moreover, these results showed that these membranes have high acidic properties. In the case of the PES membranes, this could be due to the negatively charged sulfonic acid group in the polymer structure [21].

After treatment, the zeta potential of all of the membranes changed. For the PA membranes, the zeta potential shifted toward a more positive potential at pH greater than 4.5, while for the PES/PSf membranes the zeta potential became more negative throughout the measurement range. At the pH of the TWW secondary effluent (8.78), the PA membranes (2 and 3 kDa) exhibited a high negative charge (−49.5 and −51.3 mV, respectively). This high negative charge allows adsorption of cationic compounds, such as cationic softeners, used in the factory where the TWW sample was taken. Presumably due to the polymer material (less susceptible to fouling), the thin film composite structure, and the smaller pore size of the PA membranes, the fouling compounds could not penetrate the polymer matrix; thus, adsorption of the cationic compounds occurred at the surface. Additional proof of this is provided by the SEM figures in Section 3.4.

The larger pores and looser structure of PES/PSf allow pollutants to penetrate the polymer matrix. In addition, the hydrophobic character of PSf allows the adsorption of organic compounds in the polymer matrix of the membrane. The specific adsorption between the membrane and negatively charged reactive dyes used for dyeing in the textile factory where the TWW samples were collected decreased the zeta potential of these membranes, i.e., the membranes became more negative [22,23].

### 3.3. Hydrophilicity of the Membrane

The second important parameter for measuring changes in membrane characteristics is the contact angle. Table 3 lists the average values of the contact angles of investigated membranes. All pristine UF membranes showed very similar contact angles, confirming the hydrophilic character of the membranes used, as it was below 90° [24], i.e., the membranes had intermediate properties, as the contact angle was between 45–90 °C [25]. The highest contact angle was for the 5 kDa PES/PSf membrane and the lowest was for the 10 kDa PES/PSf membrane. This difference can be attributed to the different surface modifications (which are a secret of the supplier, even if the same material is used to make the membrane itself). After secondary wastewater treatment, the contact angle increased for all membranes used, showing that fouling increases the hydrophobicity of the membrane material. No change in contact angle is observed for the 3 kDa PA membrane, whereas the largest increase (38%) is seen for the 10 kDa PES/PSf membrane. This increase can be attributed to the adsorption of hydrophobic substances in TWW on the membrane surface. Probable the main hydrophobic substances responsible for membranes hydrophobicity increase could be polysaccharides and proteins which can be released during the production process from the knitted fabric and from the drinking water used for production (fulvic and humic acids) [26]. In addition, paraffins or waxes used in the textile industry are removed by pretreatment processes for the purpose of enabling the dye to be adsorbed onto the fiber. Since this study used real TWW these hydrophobic substances can interact with membrane. According to the contact angle, the contact angle of the most fouled membranes was the largest, and this was the 10 kDa membrane.

### 3.4. Membrane Morphology

Membrane roughness as one of the morphological characteristics can show how the membrane still contributes to the retention factors of pollutants from wastewater as well as fouling problems. In this study, the membrane morphology, i.e., roughness, was investigated using AFM (Table 4 and Figure 2) and SEM (Figure 3). Table 4 shows the average roughness (*S*_a_) and Root mean square (*S*_q_) of the pristine membranes and the membranes after treatment of the TWW secondary effluent. All these results show a smooth surface of all pristine membranes, with the lowest surface roughness for the 2 kDa membrane. The surface roughness is higher for the other membranes, but they can still be considered smooth as confirmed by AFM and SEM images. After treatment, the surface roughness increased for all membranes studied, with the highest increase for the 2 and 5 kDa membranes and the lowest for the 10 kDa membrane. The 10 kDa membrane has biggest MWCO, i.e., the largest pores allowing pollutants entering the membrane matrix (pores). There was surface fouling, but we can assume that the pores were blocked first and then the surface, indicating a smaller increase in roughness. Nevertheless, the present results showed that a new layer, i.e., a fouling layer, was probably formed on the membrane surface, which is visible on the AFM (Figure 2) and SEM (Figure 3) images. Sadeghi et al. [27] found that membranes with higher surface roughness are more susceptible to fouling because the pollutants present in the wastewater are more likely to be trapped by a surface with higher roughness [28].

Fouling was confirmed by SEM images of the membrane surface and cross section (Figure 3). SEM images of all pristine membranes show homogeneous and smooth surfaces, as expected, confirming the results obtained with AFM. Only the surface of the 10 kDa pristine membrane showed unidentified particles such as dust, as only demineralized water was used to compact the membrane. A fouling layer can be seen on all images taken after treatment of the TWW secondary effluent. According to the manufacturer, the 2 and 3 kDa membranes are thin film composite membranes that have a dense membrane composition. Due to the structure of the membrane and the membrane density, the pollutants could not penetrate into the polymer matrix of the membrane and a fouling layer formed on the skin (visible on the cross-section images). The fouling layer can be seen on the images of the membrane surface as well, as this layer cracked during drying and sample preparation. The PES/PSf membranes have only a thin selective layer and a support layer with visible larger pores. Therefore, the fouling layer was not as easy to detect because the fouling mechanisms were likely cake formation, complete pore fouling, and partial pore fouling. In PES/PSf membranes, the fouling layer is not very well visible on the cross-section images. However, the fouling layer is visible on the surface images. Comparing these images with those of the pristine membrane, it can be seen that the structure of the upper layer, i.e., the fouling layer, is completely different from the surface of the pristine membrane. The structure of the fouling layer is rougher and looser. We can be sure that it is a fouling layer because it cracked during the drying of the membranes. This difference is explained by permeability in the following section.

### 3.5. Surface Structure

The FTIR spectra of pristine PA and PES/PSf membranes (Figure 4) showed typical peaks. In the fingerprint region of PA membranes, the peaks at 1487 cm^−1^ and 1601 cm^−1^ are characteristic of N–H group and C–N stretching, respectively, while PES/PSf membranes are at 1141 cm^−1^ (O–S–O stretching), 1240 cm^−1^ (C–O–C stretching), and 1585 cm^−1^ (C–C aromatic). For PA membranes, the peaks between 2850 and 3000 cm^−1^ were assigned to C–H stretching, while the broad peak at 3300 cm^−1^ represents the –NH and –OH groups.

The chemical structure of the fouled membranes was determined in comparison to the chemical structure of the pristine membranes. Although the membranes used in the study had the same transmittance peaks, they showed several different transmittance peaks compared to the pristine membranes, indicating the adhesion of pollutants on the membrane surface. For the 2 kDa membrane, new peaks at 1455 and 1545 cm^−1^ were detected, which were related to the bending vibration of CH_2_ and secondary amine, respectively. The peaks at 1045 cm^−1^ (C-O stretching (primary alcohol)) and 1711 cm^−1^ (C=O stretching) disappeared, while the peak at 1652 cm^−1^ increased and narrowed at 3000 cm^−1^. At 3 kDa, the peak at 1043 cm^−1^ and the region between 3000 and 3700 cm^−1^ were attenuated. For 5 and 10 kDa, the situation was similar, i.e., the broad peaks disappeared between 3035–3130 and lengthened between 3170–3700 cm^−1^. According to a previous study by Elcik et al. [29], the last stretching is related to the binding of reactive dyes. The change in the broad peak at 3300 cm^−1^ suggests that this membrane has abundant –OH and –NH groups on the membrane surface, as these groups can be modified by hydrophilic compounds such as reactive dyes, which is confirmed by the zeta potential.

### 3.6. Membrane Permeability

Permeate flux can provide data on membrane fouling, and is one of the simplest methods commonly used in the studies mentioned in the introduction section. In Figure 5, the permeability and normalized flux (*J*/*J*_0_) are shown for all the membranes studied during the 4.5 h treatment. From the figure, it can be seen that the 2 and 3 kDa membranes showed the lowest initial decrease, between 15% and 20% (first flux of TWW treatment compared to demineralized water). After the initial decrease, both permeability and normalized flux showed linearity, confirming that the fouling layer formed at the beginning of the treatment and only at the surface. This is additional evidence for Figure 3. Both the 5 kDa and 10 kDa membranes showed a sudden decrease in flux of about 20% in the first 15 min, however, the total decrease was 45% and 60%, respectively. This can be attributed to the larger MWCO and looser structure, which allows more accumulation of pollutants in the membrane matrix, leading to more fouling, as shown in the images from SEM (Figure 3) and our previous study [15].

## 4. Conclusions

This study investigated the surface characterization and morphology of four UF membranes after treatment of TWW secondary effluent. A difference was found between the pristine membrane and the membrane after treatment, depending on the material and the characteristics of the membrane itself. It was found that the fouling layer was much more pronounced in membranes with higher flux, which was related to MWCO. The zeta potential showed a lower zeta potential of the fouled PES/PSf, indicating a greater ability to adsorb pollutants than PA membranes. All membranes showed changes (attenuation, disappearance, lengthened) the FTIR spectra at 3000 cm^−1^, which can be attributed to the adsorption of reactive dyes from TWW. 2 and 3 kDa membranes show more compounds on the surface of the membranes and this is confirmed by the lower possibility of pore fouling. In addition, the surface roughness increases less for 10 kDa membrane because the possibility of pore fouling is greater than for the membranes with lower MWCO. Also, permeability decreases more in membranes with higher MWCO, which is also due to the larger MWCO and looser structure that allows greater accumulation of pollutants in the membrane matrix. In conclusion, this study provided an intensive characterization of UF membranes after TWW treatment, which can be used in future studies where UF membranes are used in the hybrid process. Detailed characterization showed that 2 and 3 kDa membranes could be the best option for the treatment of textile or other wastewaters.

## Figures and Tables

**Figure 1 polymers-14-02035-f001:**
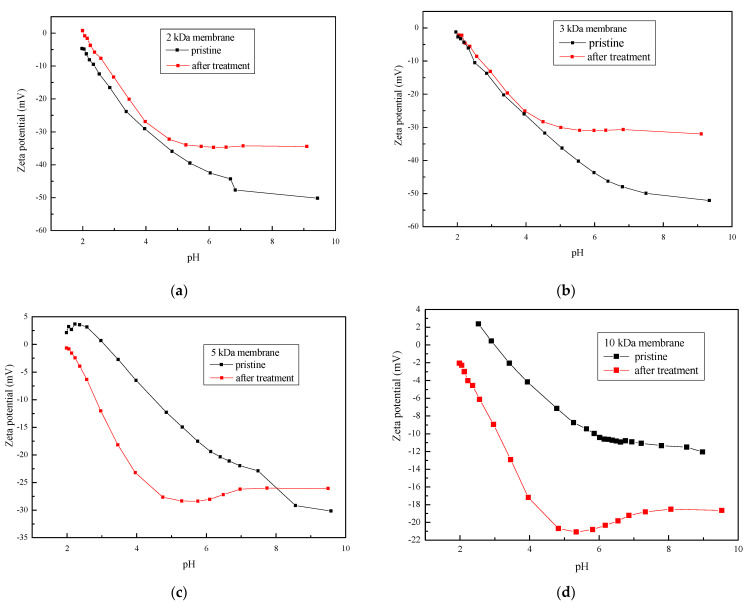
Zeta potential of pristine membranes and membranes after TWW secondary effluent treatment: (**a**) 2 kDa membrane; (**b**) 3 kDa membrane; (**c**) 5 kDa membrane; and (**d**) 10 kDa membrane.

**Figure 2 polymers-14-02035-f002:**
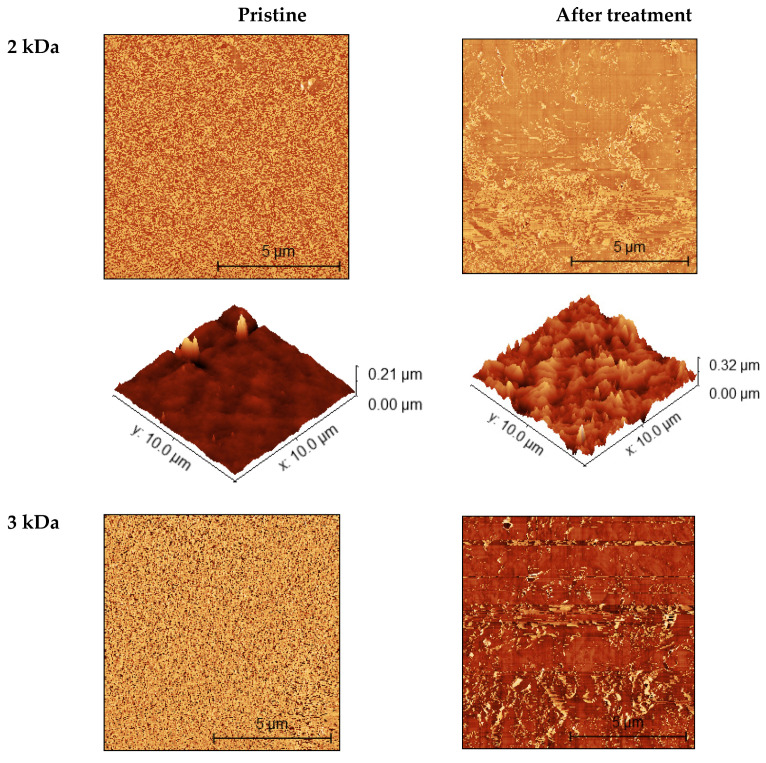
AFM images of pristine membranes and membranes after TWW secondary effluent treatment.

**Figure 3 polymers-14-02035-f003:**
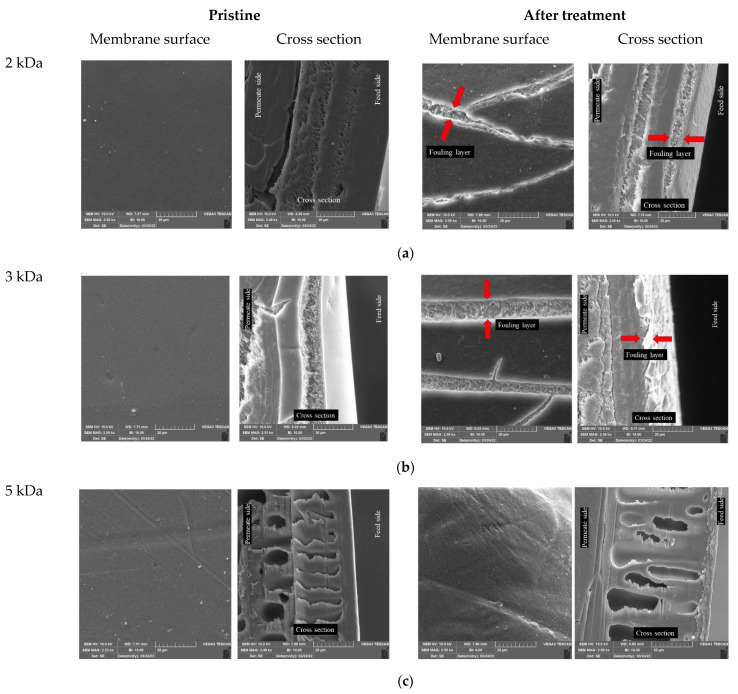
SEM images (front of the membrane and cross section) for pristine membranes and membranes after TWW secondary effluent treatment for: (**a**) 2 kDa, (**b**) 3 kDa, (**c**) 5 kDa, and (**d**) 10 kDa membrane.

**Figure 4 polymers-14-02035-f004:**
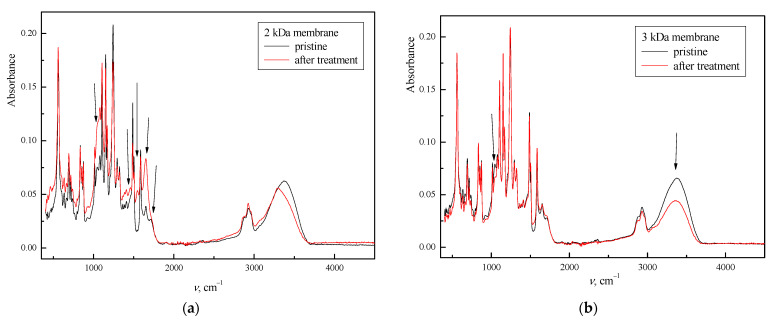
FTIR spectra of pristine membranes and membranes after TWW secondary effluent treatment: (**a**) 2 kDa, (**b**) 3 kDa, (**c**) 5 kDa, and (**d**) 10 kDa membrane.

**Figure 5 polymers-14-02035-f005:**
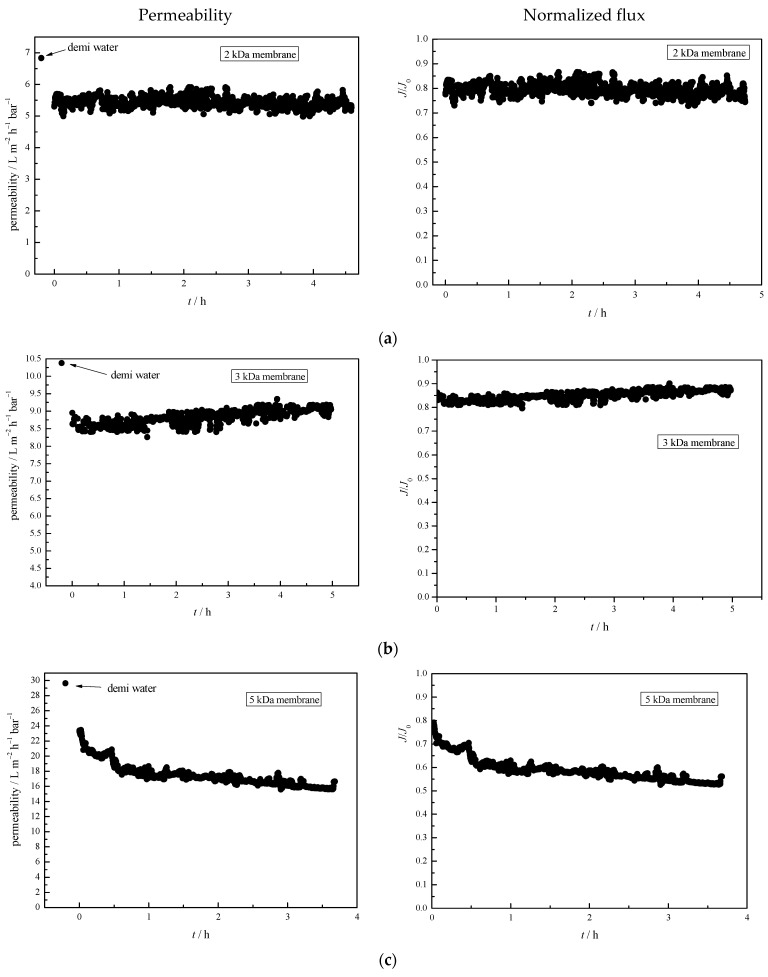
Permeability and normalized flux (*J*/*J*_0_) for investigated membranes: (**a**) 2 kDa, (**b**) 3 kDa, (**c**) 5 kDa, and (**d**) 10 kDa.

**Table 1 polymers-14-02035-t001:** Characteristics of membranes.

	GH	GK	PT	PU
Membrane	UF	UF	UF	UF
MWCO, kDa	2	3	5	10
Polymer ^1^	PA	PA	PES/PSf	PES/PSf
Maximum operating pressure, bar	27	27	10	10
Maximum operating temperature, ℃	70	70	70	70
pH range, continuous operation	1–11	1–11	1–11	1–11
Working pressure ^2^, bar	10	5	4	2

^1^ PA—polyamid, PES—Poly(ether-sulfon)/polysulfone; ^2^ Used in this study.

**Table 2 polymers-14-02035-t002:** Average characteristics of TWW secondary effluent.

Parameter	Average ^1^	SD ^2^	Parameter	Average	SD
pH	8.78	0.12	525	36	3
Conductivity, mS cm^−1^	3.33	0.10	620	30	3
Turbidity, NTU	1.37	0.77	Chloride, mg L^−1^	69.5	3.2
COD, mg L^−1^	168	22	Sulphate, mg L^−1^	1228.6	216.9
TC, mg L^−1^	319.2	2.2	Sodium, mg L^−1^	189.3	6.8
IC, mg L^−1^	218.4	4.6	Potassium, mg L^−1^	136.8	8.1
TOC, mg L^−1^	100.9	4.9	Magnesium, mg L^−1^	9.9	0.3
TN_b_, mg L^−1^	19.6	4.2	Calcium, mg L^−1^	48.7	2.8
SAC, 1 m^−1^			Hardness, mg L^−1^ CaCO_3_	162.3	7.5
436	62	3	Hardness, D	9.1	0.4

^1^ —*N* = 3; ^2^ —standard deviation.

**Table 3 polymers-14-02035-t003:** Average contact angle of pristine membranes and membranes after TWW secondary effluent treatment (*N* = 4).

Membrane		Contact Angle/°
GH	Pristine	54.61 ± 1.74
	After treatment	61.86 ± 2.82
GK	Pristine	54.10 ± 2.63
	After treatment	56.52 ± 1.80
PT	Pristine	58.06 ± 3.58
	After treatment	64.82 ± 2.92
PU	Pristine	46.19 ± 1.77
	After treatment	64.09 ± 2.93

**Table 4 polymers-14-02035-t004:** Average roughness (*S*_a_) and Root mean square (*S*_q_) of pristine membranes and membranes after TWW secondary effluent.

Membrane		*S*_a_/nm	*S*_q_/nm
2 kDa	Pristine	8.59	14.21
	After treatment	31.38	39.04
3 kDa	Pristine	18.09	23.20
	After treatment	30.41	39.07
5 kDa	Pristine	13.76	20.59
	After treatment	36.14	46.84
10 kDa	Pristine	13.91	22.09
	After treatment	19.66	28.86

## Data Availability

Not applicable.

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
