# Peer review of "Effect of Textile Wastewater Secondary Effluent on UF Membrane Characteristics"

_polymers, 2022, doi:10.3390/polym14102035_

Round 1

Reviewer 1 Report

Dolar and coworkers examined the effects of textile wastewater secondary effluent on the characteristics of ultrafiltration membrane. Commercially available membranes samples were used for this study and a number of characterization techniques were employed to study the effects of textile wast water. The study is complete, well organized and written, therefore recommend for the publication after minor revision.

  • The FTIR analysis is not consistent with the conclusion withdrawn from the spectra: All membranes show a new peak in the FTIR spectra at 3000 cm-1.  No any new transmittance peak is recognized around 3000 cm-1.
  • The discussion in the lines 190 - 194 seems are not appropriate, how hydrophobic paraffin or waxes can increase the hydrophilicity of the membrane?
  • The text in the line 164 and 172, PE membrane is not right, it should be PA membrane as described in the Table 1.

Author Response

Q1: The FTIR analysis is not consistent with the conclusion withdrawn from the spectra: All membranes show a new peak in the FTIR spectra at 3000 cm-1.  No any new transmittance peak is recognized around 3000 cm-1.

A1: Style of the FTIR spectra is changed and peaks are easier to recognized. Spectra of pristine membrane and membrane after treatment is in the same line so it can be easier to compare. Also discussion is changed.

Q2: The discussion in the lines 190 - 194 seems are not appropriate, how hydrophobic paraffin or waxes can increase the hydrophilicity of the membrane?

A2: Text is little changed. As written paraffin and waxes as also other hydrophobic substances could interact with membrane consequently changing characteristics of the membrane. In this case increasing hydrophobicity since these substances are hydrophobic. Probable the main substances were polysaccharides and proteins from the textile factory and fulvic and humic acids from drinking water. That is why their order was changed.

Q3: The text in the line 164 and 172, PE membrane is not right, it should be PA membrane as described in the Table 1.

A3: Mistakes are corrected.

Reviewer 2 Report

polymers-1683831

This paper evaluates the effect of textile wastewater secondary effluent on UF membrane characteristics. The research was done systematically, and the report is well written. A few comments below need to be addressed before it can be considered for publication.

The findings presented are not new. It is hard to find the novelty and contribution of this work to the membrane research. In the revised version, please detail the novelty of the work clearly.

Discuss the mechanisms of changes in zeta potential? What is the primary cause of the changes (pore size, polymer material, ... ?) If it was due to negatively charged dye as stated in lines 172-173, the same trend should be expected for all membranes.

The part of the membrane assigned for foulant is not correct. Those are the polymer matrices that were bent because of poor cutting of the cross-section. The cross-section image may need to be retaken to clearly show the fouling layer on the membrane surface.

Lines 260-274: Please explain the detail of the peaks in relation to the actual material contents. The membrane material and the foulant should be identified. It may be useful to explain the finding on the zeta potential.

Figure 5: Absurd finding is shown for the 3kDa membrane. Why did the permeability increase over time after a significant drop when compared to the clean water permeability? The statement in line 312 is not based on the findings. Please revise accordingly.

Author Response

Q1: The findings presented are not new. It is hard to find the novelty and contribution of this work to the membrane research. In the revised version, please detail the novelty of the work clearly.

A1: The results in this work are new in the field of treatment of textile wastewater which is top priority in the world and treatment of wastewaters. In our previous papers we showed that UF can be used for treatment and reuse of textile wastewater. Up to know there is not detailed characterization of UF membranes since mainly researchers are oriented on RO and NF membranes as a final step of the treatment. Nevertheless, if UF wants to be applied it is important to characterized changes in membranes. When we are talking about textile wastewater treatment and application of membranes this field is not yet fully investigated. Therefore, the novelty of the paper is changed in the last section of introduction.

Q2: Discuss the mechanisms of changes in zeta potential? What is the primary cause of the changes (pore size, polymer material, ... ?) If it was due to negatively charged dye as stated in lines 172-173, the same trend should be a – a expected for all membranes.

A2: Discussion is changed and was based on the membrane properties. Since this investigation was performed with real textile wastewater we can assume which pollutants were problematic in term of membrane fouling. Nevertheless, probable due to different material and pore size situation was different. Next to this it is interesting that changes were similar for the same membrane material which was pointed out in the discussion. As you mention we would expect that trend of changes should be the same or similar, but it was not. Hope that new discussion will satisfied you.

Q3: The part of the membrane assigned for foulant is not correct. Those are the polymer matrices that were bent because of poor cutting of the cross-section. The cross-section image may need to be retaken to clearly show the fouling layer on the membrane surface.

A3: Unfortunately, we cannot take new images of cross-section of the membranes since membranes are not available anymore. Fouling layer is very easily seen on membrane surface images and authors think that also it is visible on cross section images when compared to cross section of pristine membranes.

Q4: Lines 260-274: Please explain the detail of the peaks in relation to the actual material contents. The membrane material and the foulant should be identified. It may be useful to explain the finding on the zeta potential.

A4: Discussion is expanded for the membrane material and some foulants. Since it is experiment which was done with real textile wastewater it is quite hard to connect each change in FTIR spectra. Hope the expanded version will satisfy you.

Q5: Figure 5: Absurd finding is shown for the 3kDa membrane. Why did the permeability increase over time after a significant drop when compared to the clean water permeability? The statement in line 312 is not based on the findings. Please revise accordingly.

A5: Permeability of 2 and 3 kDa membranes showed quite big deviations. This can happen when working on higher pressure. It is not the case when working at lower pressures (in this experiment for 5 and 10 kDa membranes). Also, sometimes fluctuation could be if air bubble enters the apparatus and stucks at pressure vessels. Then pressure can vary and can increase. Yes, there is small increase in the permeability but according to normalized flux this increase is below 5% which is according to us not significant. At the moment we don’t have explanation for this except “problems” with apparatus.

       This is line 312: “greater than for the membranes with lower MWCO. Permeability also decreases more for”. First part of the sentence is based on the results of surface roughness and Table 4 present changes. From these results the conclusion was made. The 10 kDa membrane has biggest MWCO, i.e., largest pores allowing pollutants entering the membrane matrix (pores) and block them rather then on the surface. Of course, there was surface fouling but we can assume that firstly pore where block and then surface indicating lower increase in roughness. Second part of the sentence is part of the permeability. The overall flux decrease for 2, 3, 5, and 10 kDa membranes was 0, 0, around 35%, and around 60%, respectively. Also the initial flux decline (first flux of the TWW treatment compared to demi water flux) was the higher for 10 kDa membrane. Therefore authors can’t revise since conclusion was made according to the results.

Round 2

Reviewer 2 Report

Most of the comments in the earlier round have been addressed adequately. However, the issue related to mistakes in foulant identification from the surface and cross-section SEM images still persists. It can mislead the reader and must be addressed. Since new images can not be taken, authors can describe the findings from visual observation (maybe supported by a picture) and remove all miss leading indicators drawn on the images.

Author Response

Q1: Most of the comments in the earlier round have been addressed adequately. However, the issue related to mistakes in foulant identification from the surface and cross-section SEM images still persists. It can mislead the reader and must be addressed. Since new images can not be taken, authors can describe the findings from visual observation (maybe supported by a picture) and remove all miss leading indicators drawn on the images.

A1: Thank you for your positive comment. Indicators are removed for surface images and cross section for PT (5 kDa membrane) and at cross section for PU (10k Da) membranes. For the surface image for PU membrane indicators are left since according to us this is very well seen fouling layer. Authors agree that since cross image has some slop fouling layer cannot be easily seen. Pictures with camera or mobile phone are not useful since this is in micro range therefore it is not visible with human eye. That is why we wanted to use SEM. Some text is added to try describing findings from visual observation.